# Free-Radical Photopolymerization of Acrylonitrile Grafted onto Epoxidized Natural Rubber

**DOI:** 10.3390/polym13040660

**Published:** 2021-02-23

**Authors:** Rawdah Whba, Mohd Sukor Su’ait, Lee Tian Khoon, Salmiah Ibrahim, Nor Sabirin Mohamed, Azizan Ahmad

**Affiliations:** 1Department of Chemical Sciences, Faculty of Sciences and Technology, Universiti Kebangsaan Malaysia, Bangi 43600, Malaysia; rawdahukm31@gmail.com; 2Department of Chemistry, Faculty of Applied Sciences, Taiz University, Taiz 6803, Yemen; 3Solar Energy Research Institute (SERI), Universiti Kebangsaan Malaysia, Bangi 43600, Malaysia; mohdsukor@ukm.edu.my; 4Centre for Foundation Studies in Science, University of Malaya, Kuala Lumpur 50603, Malaysia; salmiah01@um.edu.my (S.I.); nsabirin@um.edu.my (N.S.M.); 5Research Center for Quantum Engineering Design, Faculty of Science and Technology, Universitas Airlangga, Surabaya 60286, Indonesia

**Keywords:** epoxidized natural rubber, radical polymerization, grafting, modified rubber

## Abstract

The exploitation of epoxidized natural rubber (ENR) in electrochemical applications is approaching its limits because of its poor thermo-mechanical properties. These properties could be improved by chemical and/or physical modification, including grafting and/or crosslinking techniques. In this work, acrylonitrile (ACN) has been successfully grafted onto ENR- 25 by a radical photopolymerization technique. The effect of (ACN to ENR) mole ratios on chemical structure and interaction, thermo-mechanical behaviour and that related to the viscoelastic properties of the polymer was investigated. The existence of the –C≡N functional group at the end-product of ACN-*g*-ENR is confirmed by infrared (FT-IR) and nuclear magnetic resonance (NMR) analyses. An enhanced grafting efficiency (~57%) was obtained after ACN was grafted onto the isoprene unit of ENR- 25 and showing a significant improvement in thermal stability and dielectric properties. The viscoelastic behaviour of the sample analysis showed an increase of storage modulus up to 150 × 10^3^ MPa and the temperature of glass transition (*T*_g_) was between −40 and 10 °C. The loss modulus, relaxation process, and tan delta were also described. Overall, the ACN-*g*-ENR shows a distinctive improvement in characteristics compared to ENR and can be widely used in many applications where natural rubber is used but improved thermal and mechanical properties are required. Likewise, it may also be used in electronic applications, for example, as a polymer electrolyte in batteries or supercapacitor.

## 1. Introduction

Natural rubber (NR) is an essential raw material and accounts for over 50,000 various rubber and latex-related products. Under Statistics Malaysia (2020), NR crop production in January 2020 has increased by 13.3% to 66,232 tonnes compared with previous months [1]. This value is expected to extend beyond the 2nd quarter of 2020 because of the high demand for medical gloves to fight the global pandemic, COVID-19. Amidst the pandemic situation, Malaysia’s export of NR in February (2020) surged by 13.1%, reaching 45,827 tonnes in January [2]. However, the increasing demands cannot be met and maintained in the future by relying solely on natural rubber. Therefore, it is necessary to search for more viable alternatives and innovative solutions that can be established, with superior properties, by chemical modification and cost-effectiveness procedures. Modified rubber, especially more thermally stable derivatives such as epoxide rubber, is highly desirable and may complement the sections of the market share which are currently occupied by synthetic rubbers [3]. Currently, there are two commercially available ENRs, i.e., ENR-25 and ENR-50, where 25 and 50 represent mole epoxy content in the ENR structure. ENR is a pivotal polymer owing to its unique mechanical properties and low cost [4]. The existence of epoxide groups in the ENR structure has enhanced the originality of NR properties. This is clear through its improved polarity, good strength, excellent adhesion, high hardness, compatibility blend with other polymers, and good oil and temperature resistance, as well as its low air permeability [5]. Furthermore, ENR offers a variety of potential, particularly in electrochemical applications. This is attributed to the oxygen atom, which promotes the dissociation of salt, and hence provides conductivity while maintaining the adhesion property that can promote good contact between the electrolyte layer and electrode for electronic devices [6,7].

Generally, ENR can be synthesized and obtained by a simple chemical modification. This happens through the interaction of an oxygen atom with the double bonds of the NR chains via in situ epoxidation using an acid and hydrogen peroxide [8]. However, its low *T*_g_ (~ −22 °C) and high molecular weight are considered drawbacks to its solubility and processability [4]. Moreover, all rubbers are sticky and easily deform in warmer conditions but become brittle when cold; this has greatly and adversely affected their degree of elasticity. To overcome these major drawbacks, physical or chemical modifications must be made. For example, modifications include free-radical grafting on ENR to modify it or chemically bonding the rubber chains with another functional polymer via crosslinking and vulcanization processes [9].

To improve the processability and solubility, ENR is further degraded by decreasing its molecular weight, resulting in liquid epoxidized natural rubber (LENR) [4]. Besides this, Lee et al. [10] and Rahman et al. [11,12] have reported that the molecular weight of ENR was successfully decreased via an irradiation process with a UV lamp by breaking the long chains into small chains.

Concurrent with the above objective of improving and developing new, enhanced ENRs, this work focused on grafting radical polymerization via the photopolymerization technique. Photopolymerization is a technique that initiates and propagates a polymerization reaction using light (visible or ultraviolet (UV)) to form a linear or cross-linked polymer structure. The formation of a polymer through a chain reaction caused by light is concerned with direct photoinduced polymerization reactions. As the direct formation of reactive species by light absorption on the monomer is not an efficient path, the initiation stage of the polymerization reaction requires the presence of a photoinitiator capable of producing these reactive species under light excitation. Spectral sensitivity extension (which corresponds to the best match between the light source emission spectrum and the formulation absorption spectrum) can be accomplished by using actives species: their function is to absorb the energy of the light at a wavelength where the photoinitiator is unable to work and pass the excitation. In that case, the reaction is known as a sensitized photoinduced polymerization [13]. It is a method that is widely used to convert the multifunction monomers into insoluble networks that are efficient in many industrial fields, such as films, inks, coatings, and photoresistors [14,15]. Furthermore, it is an inexpensive method, easy to handle, and applies to low-temperature and -pressure requirements [4]. This technique only includes a monomer-soluble initiator and a molecular-weight chain-transfer agent. 2,2-dimethoxy-2-phenyl-acetophenone (DMPA) was used as a photoinitiator in this work because it showed an excellent photopolymerization performance, as stated by Mishra and Yagci [16]. Several studies related to NR grafting also have been reported. Kookarinrat and Paoprasert [17] introduced the grafting of methyl methacrylate (MMA) onto NR latex to ameliorate the stability of NR by saturating the double bonds through grafting and hydrogenation reactions as a one-pot method. Several studies have also been done on MMA-*g*-ENR [4,18,19]. They found that MMA-*g*-ENR has a higher *T*_g_ (mechanically more stable), better compatibility with curing agents and phase separation and better weathering resistance.

Additionally, because of the active epoxide on ENR, many crosslinking agents have been used with ENR. At a high-energy electron, irradiation produces a free radical that reacts with the structure of the irradiated rubber and can take place either at epoxide or the -C=C- bond [20]. Yin et al. [21] has reported the introduction of the temperature responsiveness to NR with poly(N-isopropyl acrylamide) (PNIPAM) and benzoyl peroxide as a free radical cross-linker. The results displayed a straightforward strategy for crosslinking PNIPAM with NR, and the temperature responsiveness led to the creation of new, responsive, rubber-based materials and extend the range of potential applications.

In the present work, the grafting of acrylonitrile (ACN) onto ENR- 25 as a new grafting polymerization will be synthesized via free-radical photopolymerization and investigated in terms of the chemical and physical characterization. ENR- 25 was chosen because it contains fewer epoxide groups compared to ENR- 50. Eventually, this will provide more alkene double bonds in the polymeric chain for radical polymerization active sites. Previous studies have reported on “physically” crosslinking ENR, but little work has been done on “chemically” grafting ENR- 25. Nonetheless, the presence of an epoxide functional group on the polymeric chain provides more opportunities to utilize natural rubber in various fields, especially in the conductive or permeable membrane. This is because of the existence of an epoxy ring that considers a coordination position for the Li^+^ connection [22]. The ACN monomer has been selected as a grafting agent for this study. This is due to ACN’s unique properties, including its high film-ability, transparency, optical clarity, adhesive properties [23], and the highest grafting efficiency compared to vinyl monomers [24]. This research will give a new, broad view of grafting ACN onto ENR-25 via UV photopolymerization, which is anticipated to improve and enhance the thermo-mechanical behaviour, along with its dielectric properties, to prepare a desired membrane with high permittivity and flexibility.

## 2. Materials and Methods

### 2.1. Materials

Figure 1 presents the chemical structure of the epoxidized natural rubber (ENR) used in this work, with about 25% mole epoxy content (trade as ENR- 25), which was purchased from Malaysia Rubber Board (MRB) (Sungai Buloh, Selangor, Malaysia). Acrylonitrile (ACN) (C_3_H_3_N, ≥99 %, Sigma-Aldrich, Zwijndrecht, Netherlands), toluene (C_7_H_8_, 99.5%, Systerm^®^ Chemar^®^, Shah Alam, Selangor, Malaysia), 2,2-dimethoxy-2-phenyl-acetophenone (DMPA) (C_16_H_16_O_3_, 99%, Sigma-Aldrich, China China-Mainland), methanol (CH_3_OH, 99.8 %, Systerm^®^ Chemar^®^, Shah Alam, Selangor, Malaysia), basic aluminum oxide (Al_2_O_3_, 99.9%, Sigma-Aldrich, St. Louis, MO, USA), potassium carbonate (K_2_CO_3_, 98%, Sigma-Aldrich, St. Louis, Germany), and xylene (C_8_H_10_, 98.5%, R&M chemicals, Semenyih, Selangor, Malaysia). Both ENR- 25 and ACN were used with further purification.

### 2.2. Samples Preparation and Characterizations

The preparation of ACN-*g*-ENR was carried out in different stages: (1) purification of reagents, (2) synthesis of ACN-*g*-ENR via radical photopolymerization, and (3) purification of ACN-*g*-ENR products.

#### 2.2.1. Purification of Reagents

About ~20 g of ENR- 25 was cut into small portions and swelled in SCHOTT DURAN^®^ containing 400 mL toluene for 24 h. For the next 48 h, the swollen ENR- 25 was continuously stirred at 70 °C, followed by filtration through a cotton gauze pack to separate the gel from the extract. The latter was poured slowly into a beaker containing 800 mL methanol, while the solution was hand-stirred using a glass rod. The purified ENR- 25 was precipitated and stuck on the glass rod surface. The purified amount was transferred to a petri dish and dried in the fume hood before transfer to the oven, where it was dried for 24 h at 100 °C. Thereafter, the sample was further dried in a vacuum oven at 50 °C for two days until the sample achieved a constant weight. The purification of ACN was conducted using a simple column with cotton fibre, quartz sand and basic aluminium oxide to remove the inhibitor and potassium carbonate, respectively.

#### 2.2.2. UV Photopolymerization of ACN onto ENR- 25

The UV photopolymerization of ACN onto ENR- 25 was carried out at different mole ratios of (10:1), (15:1), and (20:1). For ACN_10_-g-ENR_1_ preparation, 1 g (equivalent to 0.0035 moles) of ENR- 25 was swelled in toluene (10 mL) overnight. Then, the mixture was stirred for one day, followed by adding 1.86 g (equivalent to 0.035 moles) of the ACN monomer and DMPA as a photoinitiator. The amount of DMPA ranged from 0.046 to 0.076 g based on the amount of ACN used. After that, the mixture was stirred for another 48 h before exposed to UV irradiation in a UV light box (model RS component (196-5251)) with radiation at 383 nm (λ), an intensity of 236 mW/cm^2^, and E = 3.237 eV, containing four 60 watt-lamps under a continuous flow of N_2_ gas for 3 h. The grafted product was then washed with methanol to remove excess unreacted monomer. Finally, the grafted sample was put in a vacuum oven to dry at 55 °C for 24 h. ACN_15_-g-ENR_1_, ACN_20_-g-ENR_1_, and polyacrylonitrile (PAN, control sample) were prepared by the procedures described above.

#### 2.2.3. Purification of ACN-*g*-ENR Products

Purification of ACN-*g*-ENR products was performed by cutting the grafted samples into small pieces and dissolving each ratio in 50 mL of xylene for 24 h. The mixtures were then heated at 100 °C with continuous stirring until the samples were completely dissolved. Thereafter, the sample was purified using the same method as ENR- 25 purification. Figure 2 shows synthesis of ACN-*g*-ENR-grafted samples before and after purification via photopolymerization technique at different mole ratios. A schematic representation of the proposed mechanism of grafting ACN onto ENR- 25 under UV photopolymerization is shown in Scheme 1.

#### 2.2.4. Mechanism of Reaction

The mechanism of this work was established by free-radical grafting onto ENR-25. The model of heterogeneous graft polymerization of acrylonitrile onto ENR-25 was initiated by DMPA. It should be noted that Casinos [25] and Bhattacharya et al. [26] have reported that grafting can be accomplished by “grafting *from*” or “grafting *to*” or “grafting *from-to*” approaches.

The “grafting *from*” approach is achieved by treating a substrate to generate immobilized initiators followed by polymerization (the addition of monomer to polymeric radicals followed by propagation of the grafted radical chains). However, in the “grafting *to*” approach, the radical homopolymer chains are combined with the polymeric radicals, while the “grafting *from-to*” approach is achieved by a combination reaction of radical homopolymer chains with grafted radical chains. The proposed mechanism was carried out in three main steps: (i) chain initiation; (ii) chain propagation; (iii) chain termination. In the initiation step (initiator-derived radical primary radical), the photoinitiator (DMPA) was decomposed and underwent α-cleavage under UV to form benzoyl and acetal radicals. Thereafter, the acetal radicals underwent fragmentation, yielding methyl radicals that acted as initiating radicals in the polymerization [27]. In this step, the free radical (-∙CH_3_) generated by the initiator-derived primary radical was supposed to add to the double-bond of 1,4-cis isoprene. However, because of the asymmetry of the isoprene unit of ENR- 25, the allyl radical was formed by allylic hydrogen abstraction through radical attack [4,28]. It can be said the methyl radical possessed the capability to abstract the hydrogen atom from the isoprene unit, resulting in the free radical, and became the free-radical donor to the monomer molecules [26]. For the chain propagation step, the chain radical of isoprene in ENR- 25 that was formed in the initiation step was grown by the successive addition of ACN monomers. It can be said that, in this step, a grafting mechanism was performed. For the chain termination, the growing radicals are terminated either by the combination (two free radicals react to each other) or by disproportionation (hydrogen of one radical in the beta position is transferred to another radical center to produce two molecules; one is saturated and the other one is unsaturated) [29]. It should be pointed out that photolysis has the ability to induce radical reactions leading to macro-radicals that create a linear, branch, and bridging crosslinking [30]. The important models of free radical grafting have been listed in Scheme 1 below.

According to the underlying mechanism of grafting photopolymerization, the expected final products of the grafted polymer through the chain termination were either created from two molecules of the isoprene–acrylonitrile radical—which, in this case, is called (ACN-*g*-ENR) by grafting form, as shown in step 9—or from an isoprene radical with polyacrylonitrile radical—in this case, called (ACN-*g*-ENR)—by grafting, as shown in the steps 10 and 11. It should be noted that in the grafting, there is a possibility of producing only isoprene and homopolymer, without the graft product. There is also a possibility that the isoprene–acrylonitrile radical reacts with polyacrylonitrile radical and the expected product will be (ACN-*g*-ENR) or (ACN-*g*-ENR), accompanied by the homopolymer product. In this case, it can be said the chain termination was created by from-to, as shown in steps 12 and 13, and this will be confirmed through the results and discussion.

### 2.3. Characterizations

#### 2.3.1. Fourier Transform Attenuated Total Reflection (FR-ATR)

The chemical functional group of ACN-*g*-ENR samples was assessed by Perkin-Elmer Frontiers Fourier-transform–infrared (FT-IR)/far-infrared (FIR) spectrometers, incorporated with the attenuated total reflection (ATR) accessory (Waltham, MA, USA), to indicate the grafting of ACN onto the ENR-25 structure. The analysis was done under a wavenumber from 650 to 4000 cm^−1^ with a scan resolution of 2 cm^−1^ at room temperature.

#### 2.3.2. Nuclear Magnetic Resonance (NMR)

One-dimensional (1D) (nuclear magnetic resonance (^1^H-NMR) and (^13^C-NMR)) and two dimensional (2D) (correlated spectroscopy directly coupled neighbors (COSY) and heteronuclear multiple bond correlation (HMBC)) NMR spectra of the samples were also carried out to confirm the formation of ACN-*g*-ENR and the interaction between ACN and ENR- 25. Prior to analysis, ACN-*g*-ENR (30 mg) was first dissolved in deuterated dimethyl sulfoxide (DMSO-d_6_, (0.6 mL) at 100 °C for 3 h until the samples were completely dissolved. 1D and 2D NMR spectroscopies were performed on a FT-NMR spectrometer Bruker/Advance III HD 400 MHz (Bruker, Rheinstetten, Germany) at room temperature for ENR- 25 (30 mg) and ACN (0.037 mL), separately, using deuterated chloroform (CDCl3, 0.6 mL) as a solvent.

#### 2.3.3. Gel Permeation Chromatography (GPC)

The GPC technique was used by the Waters 2414 equipment with a differential refractometer index (2414) detector and Styragel^®^ columns to record the values of average molecular weights (*M*_w_), average molecular weights (*M*_n_), and polydispersity index (PDI) of the ACN-*g*-ENR samples. A total of 5 mg of each purified sample was solubilized in 5 mL THF under 60 °C for one week to dissolve the samples completely before measurement.

#### 2.3.4. Grafting and Crosslinking Studies

The grafting yield (% GY) was calculated for the grafted products according to Equation (1) [31]
% GY = *C* − *A*/*A* × 100(1)
where *A* is the weight (g) of the taken ENR-25 and *C* is the grafted product weight. However, the grafting efficiency (% GE) of the ACN-*g*-ENR sample was determined by Soxhlet instrument by extracting the boiling sample in xylene for one day. The extracted sample was dried in an oven at 50 °C until the sample reached a constant weight. % GE was evaluated using Equation (2) [32]
% GE = (*G*_0_ − *G*_1_)/*G*_1_ × 100(2)
where *G*_0_ and *G*_1_ are the weight of the dried sample before and after extraction, respectively. The equilibrium swelling technique was used to calculate the crosslink density of ACN-*g*-ENR products [33]. Firstly, the samples were weighed to record their initial dry weight (*m*_1_). Thereafter, each sample was immersed in 50 mL toluene at room temperature for five days to obtain the swelling equilibrium. After that, the solvent was wiped off from the sample’s surfaces and the samples were then weighed to record their equilibrium weight (*m*_2_). The samples were dried in an oven at 50 °C for 24 h and reweighed to record the final dry weight (*m*_3_). The drying process was carried out until the sample accomplished a constant weight. The crosslink density was then evaluated according to Flory–Rehne, in Equation (3) [34].
υ_e_ = 1/ν [ln (1 − ν_2_) + ν_2_ + χ ν_2_^2^]/ν_2_^1/3^(3)

The ν_2_ calculated by Equation (4) and *χ* is the Flory–Huggins polymer–solvent interaction parameter, evaluated according to Equation (5) [34]
υ_2_ = *m*_3_/*ρ*/[(*m*_3_/*ρ*) + (*m*_2_ − *m*_1_)/*ρ_s_*](4)
*χ* = υ/*RT* (*δ*_1_ − *δ*_2_)(5)
where υ_e_ expresses information about the crosslink density (mole/volume), whereas ν and ν_2_ are the molar volume of the solvent and the volume fraction of the polymer in the swollen mass, respectively. *ρ*, *ρ*_s_, are the density of the polymer and solvent, respectively, *R* is the gas constant and *T* is the absolute temperature. *δ_1_* and *δ_2_* express the solubility parameter of the solvent and the polymer, respectively. However, the swelling rate was determined as in Equation (6) [35]
*S*_w_ = *w*_g_ − *w*_0_/*w*_0_(6)
where *w*_0_ and *w*_g_ are the weight of the dry and swollen samples, respectively.

#### 2.3.5. Dynamic Mechanical Analysis (DMA)

DMA was implemented to demonstrate the glass transition temperatures (*T*_g_) of the ACN-*g*-ENR products using the Perkin–Elmer DMA 800 dynamic mechanical analyzer (Wellesley, MA, USA) (1Hz and a maximum force amplitude of 1.5 N). To perform this analysis, each sample was cut into approximately 25 × 0.21 × 5 mm^3^ (length × thickness × width) with a temperature range from −60 to 50 °C, and the heating rate was 2 °C/min.

#### 2.3.6. Differential Scanning Calorimetry (DSC)

DSC analysis was held by a METTLER TOLEDO^®^ Thermal Analyzer model DSC 822e (USA) under a nitrogen flow rate of 50 mL min^−1^. Approximately ~3–5 mg of each sample was weighed and subjected to two heating–cooling cycles at a heating rate of 10 °C min^−1^ and a temperature range from −100 to 250 °C to study the thermal transitions of the prepared products.

#### 2.3.7. Thermogravimetry Analysis (TGA) and Derivation of Thermogravimetric (DTG) Analyses

TGA and DTG were performed to study the degradation temperature, mass loss, and thermal stability, as well as the compositional information, of ACN-*g*-ENR samples. TGA was carried out under the Setaram LABSYS EVO simultaneous thermal analyzer (Caluire-et-Cuire, France). The samples for TGA and DTG were carefully weighed in a range between ~ 7 and 9 mg in an aluminum crucible put in the center of the heating chamber. TGA was carried out in an inert environment under a nitrogen atmosphere with a temperature range from 30 to 600 °C and a heating rate of 10 °C min^−1^. The raw data were analyzed by the Calisto software TGA program.

#### 2.3.8. Dielectric Spectroscopy Study (DSS)

Dielectric spectroscopy (DS), an analog to the DMA test, is a classical tool adopted in order to study the dielectric behavior and relaxation process of the prepared samples. Study of the dielectric properties of the grafted ACN on an ENR structure was carried out using an impedance analyzer that allows the complex electrical magnitudes to be quantified as the impedance Z^∗^ (ω), conductivity σ^∗^ (ω), and dielectric permittivity ε^∗^ (ω). This technique is based on the measurement and subsequent evaluation of some frequency-dependent parameters, including, in particular, the complex, effective permittivity *ε*^∗^ (*ω*)
*ε*^∗^ (*ω*) = *ε*′ (*ω*) − *jε*″ (*ω*)(7)
where *ω* is the angular frequency and *ω* = 2*πf*, *j*^2^ = −1 is the imaginary constant, and *ε*′ and *ε*″ are the real and imaginary components of the complex permittivity that, respectively, regard the degree of polarization and loss mechanisms in the form of heat energy in response to an applied variable electric field. In addition, *ε*′ and *ε*″ are proportional to the energy stored or dissipation per period, respectively, in the materials. *ε*′ and *ε*″ are also called dielectric constant and dielectric loss. Moreover, the tangent of the phase angle between applied voltage and resulting currents, also termed the dissipation factor, results from [5]
tan *δ* = *ε*″/*ε*′(8)
where tan *δ* is a factor that that provides information on the loss of energy and is highly based on the physical conditions of the insulating material. T The relaxation time, τ can be calculated from Equation (9) by using the maximum frequency, *f*_m_ value obtained from the graph of tan *δ* versus frequency) [36]
*ω* × *τ* = 2π*f*_m_ × *τ* = 1(9)
where *ω* express the angular frequency of the applied field and *τ* is Debye relaxation time. In this work, DSS measurements were performed via electrochemical impedance spectroscopy (EIS) with a single-channel VersaSTAT4 Schlumberger Instrument (Berwyn, PA, USA). The samples for DSS measurements were films of approximately 16–18 mm in thickness, pre-pared by in-suit UV curing. The samples were subjected to the frequency region (0.1 Hz–1 MHz) of 100 mV amplitude and room temperature. Each film, after being cut into a circular shape, was sandwiched between two identical stainless-steel ion blocking electrodes with a surface contact area of 1.76 cm^2^ after calculating the average thickness between them (0.016–0.018 cm). From the impedance data, the real and imaginary permittivity, respectively, (*ε*_r_), (*ε*_i_), real electrical modulus (*M*_r_), imaginary electrical modulus (*M*_i_), dielectric loss tangent (tan *δ*), and relaxation time (*τ*) for ACN-*g*-ENR samples were evaluated according to the equations reported by Woo et al. [36] and Basri et al. [37].

## 3. Results and Discussion

### 3.1. Chemical interaction

FTIR was performed to identify the constituent interaction of ACN onto ENR- 25. The absorbance peaks at 2962, 2924, and 2855 cm^−1^ in (Figure 3a), which refer to –CH symmetric stretching (*υ*_s_–CH_3_), –CH asymmetric stretching (*υ*_as_–CH_2_), and –CH symmetric stretching (*υ*_s_–CH_2_) in the ENR- 25 structure, were not noticeably affected after grafting.

It can be noticed in the FTIR spectra below (Figure 3b) that all products showed an additional peak at 2242 cm^−1^. This peak is owing to the stretching vibrations of –C≡N, which consider the main functional group in the ACN structure. Besides, the peak in the –C≡N stretching was of medium intensity and observed at 2230 cm^−1^ for ACN corroborates with the grafted ENR-25 samples at 2242 cm^−1^. Compared with ENR- 25, the existence of this peak proves that the grafting of ACN occurred successfully onto the ENR- 25 backbone. It should be pointed out the –C=C– stretching in ENR- 25 at 1664 cm^−1^ was shifted to a high wavenumber after grafting. This indicated that the chain initiation within the proposed mechanism was carried out via free radical allylic hydrogen not an addition to vinyl group –C=C–, as shown in Scheme 1, step 2. Furthermore, three new peaks were observed at 1710, 1717, and 1723 cm^−1^ for ACN_10_-*g*-ENR_1_, ACN_15_-*g*-ENR_1_, and ACN_20_-*g*-ENR_1_, respectively. This is attributed to the double bond in –C=N stretch as a result of resonance in ACN monomer [38], which appeared at 1713 and 1728 cm^−1^ for ACN and PAN, respectively. It can be deduced that the grafting was achieved by grafting from-to, as shown in (Scheme 1, Step 13). 

It should be pointed out that in-plane bending (scissoring) (*δ*_s_–CH_2_) and out-of-plane bending (wagging) (*ω*–CH_2_) in (Figure 3c) were shifted to lower wavenumbers of 1447 and 1377 cm^−1^, respectively, for all grafted samples compared to ENR- 25, ACN, and PAN. The in-plane bending (rocking) (*ρ*–CH_2_) in (Figure 3d), was found at 751 and 753 cm^−1^ for ACN_15_-*g*-ENR_1_ and ACN_20_-*g*-ENR_1_, respectively. This indicates that there is a change in the ENR- 25 structure after grafting. Moreover, the FTIR spectra revealed that the absorption peak at 835 cm^−1^, belonging to =C–H wagging, shifted to a lower wavenumber of ~ 832 cm^−1^ for all grafted products. This is a result of the involvement of double bonds during the chain scission, [4] as shown in the proposed mechanism (Scheme 1, step 2), and will be explained in the GPC section.

It should also be noted that the oxirane ring at 870 cm^−1^ for ENR- 25 does not exhibit any substantial change, which implies that the oxirane group was not included in the chain scission reaction [4]. This indicated the grafting of ACN onto ENR- 25 successfully occurred on –C=C– in the rubber chain instead of the oxirane ring. The analyses below also indicate the formation of grafting of ACN onto the ENR-25 backbone. Appendix A shows the FTIR’s wavenumber assignments of ENR- 25, ACN, PAN, and ACN-*g*-ENR products at different mole ratios.

1D and 2D NMRs were carried out to test the structure of ENR- 25 after the reaction with ACN and confirm the chemical interaction between ACN and ENR- 25. 1D NMR spectra for ACN, PAN, and ACN_10_-*g*-ENR_1_, and ACN_20_-*g*-ENR_1_ are presented in Appendix A, and 2D NMR spectra of PAN and ACN_10_-*g*-ENR_1_, and ACN_20_-*g*-ENR_1_ are given in Appendix A. The ^1^H-NMR of the purified ENR- 25 (400 MHz, CDCl_3_, *δ* (ppm)) in (Figure 4a) exhibited signals similar to the previous studies [39,40]: *δ* 1.29 ppm (-CH_2_-C**H**_3_COCH), *δ* 1.57 ppm (C**H**_2_–CH_3_CO), *δ* 1.70 ppm (–CH_2_–C**H**_3_C=CH), *δ* 2.20 ppm (–CH_3_C=CH-C**H**_2_), *δ* 2.06 ppm (–CH_3_COCH–C**H**_2_–), *δ* 2.72 ppm (–CH_3_COC**H**–), *δ* 5.17 ppm (–CH_3_C=C**H**–), and *δ* 7.27 ppm (the residual of CDCl_3_ solvent).

Figure 4a shows that the ^1^H-NMR chemical shift at *δ* 1.29 ppm is assigned to the methyl protons bound to the epoxidized isoprene unit, the shift at *δ* 2.72 ppm to the methine protons of the epoxidized isoprene unit, and the shift at *δ* 1.57 and *δ* 2.06 ppm to the methylene protons that bound to the epoxidized isoprene unit. However, the chemical shifts observed at *δ* 1.70, *δ* 2.20, and *δ* 5.17 ppm represent the methyl, methylene (allylic proton), and methine (vinylic proton) bonded to the isoprene unit. The methylene protons as shown in the structure (–C**H**_2_–CH_3_C=CH) were overlapped with the methylene protons bonded to the epoxidized isoprene unit. The overlapping signal may be attributed to the methylene protons of the ENR-25 chain [40]. Figure 4b and Appendix A show the ^1^H-NMR (400 MHz, DMSO-d_6_, *δ* (ppm)) for all samples of ACN-*g*-ENR. It was observed that the olefinic methine proton at 5.17 ppm did not disappear after grafting. Besides this, a signal at *δ* 2.20 ppm was assigned to the hydrogen allylic group at position **b,** as shown in the structure ENR 25 in ^1^H-NMR spectrum. This indicates that chain initiation was carried out via free radical allylic hydrogen, as shown in the proposed mechanism and discussed earlier. Furthermore, the signal at 1.30 ppm was not affected after grafting. As such, it can be deduced that the grafting of ACN onto the backbone ENR-25 did not occur via the ring-opening reaction of the protonated epoxidized ring, and also the oxirane group was not included in the chain scission reaction [4]. Additional signals appeared within *δ* 1.5 and *δ* 3.16–3.39 ppm, related to (-C**H**_2_-C**H**-CN), respectively. This proves that ACN was grafted on the ENR backbone. It should be noted that protons of ACN are assigned at *δ* 5.7–6.3 ppm, as shown in Appendix A. However, the vinyl group disappeared when the ACN converted to the monomer radical. In addition, this signal also appeared in ^1^HNMR of the PAN spectrum in Appendix A. It can be said that the grafting was achieved by grafting *from-to*, in step 13. In addition, it was observed that the chemical shift in -CH_3_ and –CH_2_ bound to an isoprene unit shifted to a lower chemical shift due to the change in the structure of ENR after grafting.

The ^13^C-NMR spectrum of the purified ENR-25 (400 MHz, CDCl_3_, *δ* (ppm)) above is shown in (Figure 4c). The signals within *δ* 22.3 and 33.2 ppm represent methyl carbons at C^5^ and C^10^ and the doublets of methylene carbons of C^3^, C^4^, C^8^ and C^9^. The signals at 60 and 64 ppm represent carbons of the epoxy ring C^6^ and C^7^, respectively. The signal at 77 ppm is assigned to the residual solvent of CDCl_3_, and 125.2 ppm and 135.6 are represented by C^1^ and C^2^ from the isoprene unit, respectively. It should be pointed out that the methylene carbons of C^4^ and C^9^ were located in the lower frequency region (upfield) compared to the methylene carbons of C^3^ and C^8^. This is owing to the methine carbons of C^2^ and C^7^ that, respective to the vicinal methylene carbons of C^4^ and C^9^, have greater electron density compared to the quaternary carbons of C^1^ and C^6^, respective to the vicinal methylene carbons of C^3^ and C^8^. Therefore, they shielded the methylene carbons C^4^ and C^9^ more [40]. The spectrums of ^13^C-NMR (400 MHz, CDCl_3_, *δ* (ppm)) for ACN and (400 MHz, DMSO-d_6_, *δ* (ppm)) for PAN in Appendix A show signals at 116 ppm for ACN and 119 ppm for PAN. These signals are ascribed to (–C≡N) and appeared in all ratios of ACN-*g*-ENR in Figure 4d and Appendix A. Moreover, the signal at *δ* 125–136 ppm for (–C=C–) shifted to a lower chemical shift, ascribed to the change in the structure after grafting. Besides, the chemical shift at C^at^, C^7^, and C^10^ did not detect after grafting. This indicates that the oxirane ring was not included in the chain scission. The signals between *δ* 20 and 40 ppm are related to the carbons of (-**C**H_2_ and –**C**H) in the PAN structure; however, the signal at *δ* 40 ppm is related to the residual of DMSO-d_6_.

2D NMR was done to study the correlation of ^1^H-^1^H and ^1^H-^13^C after grafting of ACN onto the ENR- 25 structure. The COSY spectrum of ENR- 25 in (Figure 5a) shows that the methine proton of isoprene C^2^ at *δ* 5.17 ppm was clearly correlated to the signal at *δ* 2.20 ppm (the methylene protons, C^4^) and the signal at *δ* 1.70 ppm of the methyl group through allylic (four-bonds). However, the spectrum of the grafted sample in (Figure 5b) shows only the signal at *δ* 1.9–3.2 ppm, related to ACN protons. For the HMBC spectrum in Figure 5c,d, it was observed in the ENR- 25 spectrum, as reported by Hamzah et al. [40], that the signal at *δ* 5.17 ppm correlates to (C^3^, C^4^, and C^5^), the signal at *δ* 1.30 ppm correlates to (C^7^and C^8^), at *δ* 1.70 pm correlates to (C^2^ and C^3^), the signal at *δ* 1.57 ppm correlates to (C^6^, C^7^, and C^10^), the signal at *δ* 2.06 ppm correlates to (C^6^and C^7^), the signal at *δ* 2.20 ppm correlates to (C^1^and C^2^), the signal at *δ* 2.72 ppm correlates to (C^8^, C^9^, and C^10^), and that at H, belonging to C^3^, correlates to (C^1^, C^2^, and C^5^). However, all ACN-*g*-ENR samples exhibit a new signal at *δ* 1.9 and *δ* 3.1 ppm, correlating to C at *δ* 120 ppm.

### 3.2. Molecular Weight Distribution Measurements (MWDM)

GPC was carried out to determine the values of the average molecular weights (*M*_w_), number average molecular weights (*M*_n_), and polydispersity index (PDI) of the grafted products, as shown in Table 1 below. The values of *M*_w_ for the purified ENR- 25, liquid epoxidized natural rubber (LENR 25), and PAN (under UV as control samples) were recorded at ~ 1287,538, ~ 51,540, and ~77,601 g mole^−1^, respectively. However, it was found that the value of *M*_w_ was decreased to ~ 171,064, ~ 114,657, and ~ 252,786 g mole^−1^ for ACN_10_-*g*-ENR_1_, ACN_15_-*g*-ENR_1_, and ACN_20_-*g*-ENR_1_, respectively, compared to the purified ENR- 25. This decrease is due to the degradation of ENR-25 under UV, which contributed to the decrease in the *M*_w_, as reported by Rooshenass et al. [4], Lee et al. [10], and Rahman et al., [11,12]. In addition, it can be deduced that the ENR-25 was degraded by looking at LENR as a control sample; it was observed that the *M*_w_ decreased under UV compared to ENR- 25. Moreover, the decrease in *M*_w_ is due to the chain scission of ENR-25, as shown in Scheme 1, step 2 and step 13 when ACN created a radical molecule that induced the macromolecular to produce chain scission. As a result, the underlining mechanism of the grafting process can be said to have been carried out by the *from-to* mechanism, as shown in (Scheme 1, step 13).

Although no unique relation was found by Azizi et al. [41] for the variation in grafting and cross-linking performance as a function of the MWDM, they have reported increasing the grafting level and gel content with increasing the values of M_n_ and M_w_ increased. This is owing to several long chains that are present in the synthesized polymer. Contrary, in this work, the lowest M_w_ obtained the highest grafting. Furthermore, the highest Mw was obtained in ACN_20_-*g*-ENR_1_ compared to the other mole ratios. This is ascribed to the grafting in this ratio, which has led to increasing the chain length rather than increasing the reaction site [42]. However, ACN_15_-*g*-ENR_1_ was given the lowest M_w_ because of the increased reaction site, which led to an increase in its grafting efficiency, as will be discussed in the section on grafting efficiency and cross-linking density. It should also be noted that higher PDI values were obtained for the grafted samples. Such a case is not uncommon since free-radical polymerization often performs poorly in controlling M_w_ and PDI. This is due to the chain scission that was caused by the disproportion process. Additionally, the molecular-weight parameters, heterogeneity in cross-linking, network formation, and chain length lead to more random arrangements and increased PDI. It can also be seen that the type of polymerization and the photoinitiator play a role in increasing or decreasing the PDI; this was clear from the comparison between this work and that done by Li et al. [43]. The PDI of PAN in this work, as a control, was found to be 4.9; however, the PAN synthesized by living polymerization was 1.13. Autoscaled Chromatogram GPC of PAN, LENR 25, ENR- 25, ACN-*g*-ENR at various mole ratios is shown in Appendix A.

### 3.3. Grafting Efficiency and Crosslinking Density

The grafting yield (% GY) and grafting efficiency (% GE) of the grafted products ACN-*g*-ENR were directly proportional to the addition of an ACN monomer until they achieved the maximum value at ACN_15_-*g*-ENR_1_. It was noted that % GY and % GE increased dramatically with the increase in ACN amount, and decreased thereafter, with a further increase in ACN, as explained in Appendix A. It should be noted that the sample ACN_15_-*g*-ENR_1_ achieved the highest % GY and % GE of 64.05 % and 56.59 %, respectively. This may be related to the extent of grafting that is controlled by the amount of ACN diffusion and the reaction medium reaching the ENR-25 structure, as discussed earlier. However, after the optimum ratio of ACN_15_-*g*-ENR_1_, both % GY and % GE decreased. This could be attributed to the tendency of PAN homopolymer formation as ACN content to increase onto the ENR-25 backbone, thereby increasing the reaction medium viscosity. Eventually, this restricts the motion of the ACN monomers reaching the active sites at the ENR-25 backbones, which directly reduces the rates of grafting [30], as shown in Scheme 1, step 13.

The equilibrium swelling revealed the production of a crosslinking reaction during the grafting of ACN onto ENR-25. This was clear when ENR-25 was immersed in toluene; it dissolved absolutely after 5 days. However, the grafted samples did not completely dissolve in toluene in the same period of time. This could be due to the presence of macroradicals that create the crosslinking during grafting. Appendix A explained that, when the amount of ACN increased, the swelling ratio was decreased as a result of crosslink density increasing when the amount of ACN increased. Table 2 shows the % solubility of the grafted products (for non-purified samples) using different organic solvents.

### 3.4. Thermal Analyses

DMA analysis was performed to elucidate the changes in storage modulus (*E*). In general, the change in storage modulus, *E*, is a direct illustration of the intermolecular interaction between the polymers. The storage modulus curve of all grafted samples is categorized into three regions. First, a large value of storage modulus (19,779 to 132,353 MPa) was noticed at a very low temperature, which is a glassy state. Second, a drastic decrease in *E* is due to the glass transition region. Third, the *E* showed a plateau curve, namely, a rubbery state [5]. At a low temperature, the *E* did not appear to be affected, as the deformation was primarily elastic due to the decrease in molecular motion [44]. The E value slightly declined between −60 and −40 °C, then decreased rapidly after −40 °C until a rubbery plateau was reached. The samples exhibited closed *E* at 25 °C in the range of 55–56,222 MPa. The tan *δ* peak achieved its maximum peak by the reduction in *E* in the transition region. The behavior is associated with the transition of the glassy–rubbery phase, which is ascribed to the motion of the micro-Brownian of the main chain, also called α-relaxation [45]. The value of *T*_g_, is given by the highest peak (α-relaxation) of tan *δ* vs. temperature curve, was found at −12 and −16.63 °C, with tan *δ* 0.37 and 1.05 for the ACN_10_-*g*-ENR_1_ and ACN_15_-*g*-ENR_1_, respectively.

However, ACN_20_-*g*-ENR_1_ showed α-relaxation, *T*_g_ at 13 °C, with another β-relaxation at −16.16 °C with tan *δ* 0.34. It was noted that the transition area of *E* and the position of tan *δ* peak shifted towards the higher temperature, with the increasing mole ratio of ACN to ENR-25, as explained in Figure 6.

By comparing NR, ENR- 25, ENR- 50, Acrylonitrile Butadiene Rubber (NBR) and Acrylonitrile Butadiene Styrene (ABS) were reported by Salaeh [5] and Ramesan and Alex [46]; ACN-*g*-ENR films showed a large value of *E* in the rubbery area com-pared to neat NR, ENR- 25, ENR- 50, NBR and ABS, as shown in Table 3. Ramesan and Alex, [44] used Aparene N553 NS, with 34% bound acrylonitrile content, which is substantially higher compared to this work; however, its mechanical properties were considerably poor. In our case, ACN-*g*-ENR has better mechanical properties. This is evidenced by their higher degree of crystallinity and the strong interaction between ACN and ENR- 25, which generated transient cross-linking. Thus, it displays increased *T*_g_ values, as demonstrated in the DSC section. Nevertheless, the tan *δ* values of ACN-*g*-ENR films decreased with increasing ACN content. This is ascribed to the increase in interfacial bonding [44] as a result of the increase in the crystallinity of grafted products. The DMA result showed that the ratio of ACN_15_-*g*-ENR_1_ is more flexible, due to it having the highest value of tan *δ*. The ratio of ACN_20_-*g*-ENR_1_ showed the highest crosslinking, as explained earlier in crosslinking studies. Thus, it is deduced that the grafting of ACN onto the ENR- 25 matrix was suitable to alter ENR, and then can be used to prepare membranes or composites with high dielectric-thermo-mechanical properties.

DSC thermograms (second run curves) of the PAN homopolymer, with ENR- 25 as control samples, and ACN-*g*-ENR at different mole ratios, are illustrated in Figure 7. The *T*_g_ for PAN and ENR- 25 was 97 and −44 °C, respectively. This value is very close to the previously reported *T*_g_ of PAN (100 °C), [47], and ENR- 25 (−40 °C) [7]. However, a sharp exothermic peak between about 240 and 350 °C in PAN homopolymer is not exhibited in the thermogram, as reported by Preta et al. [48] and Fleming et al. [49], due to the measurement being carried out between −100 and 250 °C. This peak was due to dehydrogenation, oxidation, crosslinking, and cyclization reactions that were initiated by the free-radical mechanism. However, thermogravimetric analysis shows signs of this peak, which will be explained in the TGA analysis section. The thermal stability of ACN-*g*-ENR samples exhibited a notable enhancement due to the grafting reaction. As summarized in Table 4, the *T*_g_ values increase gradually from −44 to −34.39 °C. The presence of another *T*_g_ was observed at ~103, ~103, and 111 °C for ACN_10_-*g*-ENR_1_, ACN_15_-*g*-ENR_1_, and ACN_20_-*g*-ENR_1_, respectively. This is due to the presence of a homopolymer of PAN that was cross-linked beside the grafted product, as shown in Scheme 1, step 13. In addition, some peaks were observed at 175 and 205 °C for the ratio ACN_10_-*g*-ENR_1_, a peak at 180 °C for the ratio ACN_15_-*g*-ENR_1_, and two peaks at 177 and 209 °C for the ratio ACN_20_-*g*-ENR_1_. These new peaks also suggest the presence of cross-linking via UV curing. As shown in Figure 7, the grafting of ACN onto ENR-25 causes an increase in *T*_g_. As mentioned earlier in the GPC part, this is due to chain scission, which causes the polymer chain to become smaller and leads to transient cross-linking. Table 4 summarizes the main data for the DSC thermal analysis thermograms.

Thermogravimetric analysis (TGA) and derivation of thermogravimetric (DTG) were executed to investigate the thermal decomposition of ENR- 25, PAN, and grafting products. Figure 8 illustrates the thermal degradation characteristics of ENR- 25, PAN, and ACN-*g*-ENR films as a function of % weight loss with temperature. However, about 9–13% weight loss was noticed for all samples at the temperature range of 50–100 °C. This is due to the evaporation of some remaining solvent trapped inside the samples. All samples exhibited only one thermal degradation stage in the temperature range of 50–600 °C. In the temperature range of 100–300 °C, peaks were detected for all grafted products. These peaks are associated with the dehydrogenation, oxidation, and crosslinking of PAN, as explained in the DSC section. Thermal degradation values higher than neat ENR- 25 were also observed for the grafted samples. This indicates the grafting of ACN onto the ENR- backbone, which is a popular behaviour because of the presence of ACN, which exhibits high thermal stability, as most studies have observed. The values of *T*_max_ are higher for the grafted samples compared to ENR-25, proving that the grafting of ACN onto ENR- 25 influences the onset and outset of the grafted thermal degradation [50]. Furthermore, the temperatures of the maximum weight loss rate of ENR- 25 and ACN-*g*-ENR graft samples are different, indicating that grafting ACN onto the ENR-25 backbone changes the maximum weight loss degradation behaviour of the ENR-25 matrix [4]. The main weight loss is due to the chain scission of the grafted ACN and the total decomposition of the rubber matrices. Although there is a reduction in the *M*_wt_ of ACN-*g*-ENR samples, as explained in the GPC section, the grafted samples remained thermally stable, as shown by the TGA results. From these degradations, it can be deduced that the graft products exhibited high thermal stability. This shift can be explained by assuming that the produced grafting and cross-linking of ACN onto ENR-25 improved the thermal stability [8,9,32,51], as shown in Figure 8, which can be used to prepare a membrane with high thermal stability.

The TGA result also showed that the ACN_15_-*g*-ENR_1_ is more stable compared to ACN_10_-*g*-ENR_1_ and ACN_20_-*g*-ENR_1_. This is due to this ratio having the highest % GY and % GE, as explained earlier in the grafting studies. It should be pointed out that the residue fraction was high at a temperature of over 500 °C: around ~ 23.86 %, 40.77 %, and 36.53 % for ACN_10_-*g*-ENR_1_, ACN_15_-*g*-ENR_1_, and ACN_20_-*g*-ENR_1_, respectively. This could be due to the scission of the main chains. Nevertheless, the increase in ACN-*g*-ENR residue with ACN loading is due to undecomposed can, as a temperature of 600 °C is not sufficient to completely decompose the ACN. According to Salles et al., the complete decomposition of PAN with complete removal of carbon occurs at 1000 °C [52]. Table 4 lists *T*_d_, *T*_max_, and wt. % for ENR- 25, PAN, and ACN-*g*-ENR products.

### 3.5. Dielectric Spectroscopy Study

In this work, the dielectric spectroscopy study (DSS) was demonstrated at room temperature and applied to address the influence of grafting on the structural relaxation of the ENR elastomer, as shown in Figure 9. In addition, this aimed to increase understanding the properties of the grafted product and provide a well-characterized polymeric matrix compared to the previous literature. The real permittivity, *ε*_r_, and imaginary permittivity, *ε*_i_, as functions of the frequency for the grafting ACN on the structure ENR at different mole ratios, were plotted in Figure 9b,c. It was observed that both *ε_r_* and *ε_i_* decreased with frequency due to the orientation of the dipole and the polarization of the displacement, which were unable to follow the changes in the alternating electrical field at high frequency. At high frequencies, the dipoles do not have time to align before the field changes direction, thus resulting in a decrease in charge accumulation. On the contrary, at low frequencies, the dipoles have sufficient time to align with the field before it changes direction. This can be clearly seen as both *ε_r_* and *ε_i_* slowly start to decrease from their initial values, followed by a rapid decrement with increasing frequency, before reaching a nearly constant value. The rapid decrease in *ε_r_* and *ε_i_* values with increasing frequency can be associated with the inability of dipoles in the system to rotate rapidly, leading to a lag between the frequency of oscillating dipole and that of the applied field [53].

A comparison of our work (*ε_r_* and *ε_i_*) and previous work by Salaeh [5] is summarized in Appendix A. Overall, the values of *ε_r_* and *ε_i_* for ACN-*g*-ENR were higher compared to pure ENR 25 (*ε_r_* = 3.76 and *ε_i_* = 0.03). The *ε_r_* and *ε_i_* values for both ACN_15_-*g*-ENR_1_ and ACN_20_-*g*-ENR_1_ are ~ 8.04 and 10.40 and ~ 6.71 and 6.87, respectively. This is attributed to ACN properties, which also evidenced that ACN is successfully embedded into the ENR- 25 backbones. Additionally, the high value of *ε_r_* and *ε_i_* in the sample indicates a higher amount of charge carriers and energy losses, respectively [36,54], leading to high charge mobility when the polarity of the electric field changes quickly [36,54,55]. Overall, ACN_15_-*g*-ENR_1_ represents the highest dielectric values and an improvement over ENR samples, possibly indicating that it possessed the highest grafting efficiency, as discussed earlier, and has the lowest *T*_g_, as reported in the DSC section.

Besides the permittivity study, the electric modulus can be used to describe the relaxation phenomenon in the material [56]. In Figure 9d,e, two relaxation peaks were observed. The two peaks are attributed to the *α* and *β* –relaxation process, which generally contributes a segmental motion [57] and crosslinking of the formed homopolymer PAN. Here, the observed relaxation behaviour is the result of the combination of two polarization mechanisms, which depend on the physical movement of the charges responsible for the dipoles and on the length of time required for their displacement. One mechanism originates from the dielectric properties of the ENR of the polar epoxy group, and the other is the ACN of the polar nitrile group. In general, the ratio ACN_20_-*g*-ENR_1_ gave the highest value of *M*_r_ and *M*_i_ compared to ACN_15_-*g*-ENR_1_. This is ascribed to the higher crosslinking density in the ACN_20_-*g*-ENR_1_, as shown in Appendix A. As reported by Zulkifli, [53], both α and β relaxation increase with the increase in the amount of cross-linking reaction. However, except for the ACN_10_-*g*-ENR_1_, there are no real significant changes in electric modulus above the ratio of 10:1 (ACN: ENR).

For the tan delta (Tan *δ*), the relaxation time can be estimated using *ω* × *τ* = 2π*f*_m_ × *τ* = 1 equation from the maximum tan *δ* vs. frequency curves, as mentioned earlier. The maximum tan *δ* (also known as maximum dielectric loss tangent) is a time constant, which represents the relaxation time associated with the change diffusion process in the sample and is correlated with the plateau of the real part of the conductivity [58]. It can be seen in Figure 9f that tan *δ* increased with increasing values of frequency, to a maximum, and then decreased. At the lower-frequency region, the tan *δ* increased due to the dominance of the Ohmic (active) compared to the capacitive (reactive), while the reverse trend is observed at the high-frequency region [59]. From the results, the value of the obtained *τ* is 0.023 and 0.039 s for ACN_15_-*g*-ENR_1_ and ACN_20_-*g*-ENR_1_, respectively. The relaxation time was calculated from the frequency, *f*_m_ corresponding to the peak according to the equation reported by Woo et al. [36]. The ACN_15_-*g*-ENR_1_ sample exhibited the shortest relaxation time, which indicated that ion dynamics are strongly governed by the segmental motion of the polymer chain and increase the rate of segmental dynamics (*τ*_tan__δ_), which increases the mobility of ions in the faster-relaxing media [60]. However, no peak is observed for ACN_10_-*g*-ENR_1_.

It can be concluded that the results of dielectric analysis illustrated that the ACN_15_-*g*-ENR_1_ sample is suitable to be a host polymer for electrochemical applications, owing to it showing an enhancement in the dielectric behavior compared to the other samples. Appendix A summarizes the dielectro-thermo-mechanical proper-ties of NR, ENR- 25, ENR- 50, and NBR from the previous studies, compared to ACN-*g*-ENR synthesized at various mole ratios [61,62,63,64,65,66,67,68,69,70].

## 4. Conclusions

The grafting of an ACN monomer onto ENR- 25 material, initiated by a DMPA initiator, was successfully conducted by free-radical polymerization at a different mole ratio of ACN to ENR- 25 by grafting *from-to*. The optimum result for grafting was found for ACN_15_-g-ENR_1_, with the optimum % GY = 64.05% and % GY = 56.59%. However, at a higher ratio of monomer, *M*_w_ was increased and PDI decreased. The % GE and crosslinking density were dramatically increased with an increase in ACN moles. However, after obtaining the optimum monomer addition for ACN_15_-*g*-ENR_1_, both % GY and % GE decreased. Thermal analysis showed a high *T*_g_ value after the grafting of ACN onto ENR 25, due to lower chain mobility and an improvement in the overall thermal stability. The results of dielectric analysis illustrated that ACN-*g*-ENR is suitable as a host polymer for electrochemical applications owing to its higher mechanical properties and dielectric permittivity. The results show that the grafting of ACN onto ENR- 25 decreased the molecular weight and increased *T*_g_ and the thermal stability. Overall, it may be said that, by comparing NR and ENR-25 to ACN-*g*-ENR, ACN-*g*-ENR will introduce the desired properties for ENR-25, which increases its applications in the various applied fields. This may be the first step to achieving an economical production, with a simple and speedy method, for an industrial manufacturing line.

## Data Availability

The data presented in this study are available on request from the corresponding author. The data are not publicly available due to privacy restrictions.

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
