# Peer review of "Free-Radical Photopolymerization of Acrylonitrile Grafted onto Epoxidized Natural Rubber"

_polymers, 2021, doi:10.3390/polym13040660_

Round 1
Reviewer 1 Report
The article titled “Free-radical Photopolymerization of Acrylonitrile Grafted onto 2 Epoxidized Natural Rubber”. This is an interesting study and has importance to the polymer scientists. However, a few of the concerns are listed below.
- In abstract, line no. 24, authors have mentioned “a high grafting efficiency (~57%)”, this is not so high, hence, the phrase could be “enhanced grafting efficiency”.
- Why better PDI (Table 1) could not be achieved? A sentence explaining this could be inserted.
- The supplementary data could not be seen as the data were not available online. Did the authors missed to upload those? Are the GPC plot(s) shown in the supplementary data file?
- Can the energy obtained from the UV light that was used for the photopolymerization reaction be included in the manuscript?
Author Response
Hi,
A good day to you!
We appreciated your valuable comments and feedback.
Kindly please find the attached revision and our feedbacks to your comments.
Thank you
Best regards,
Lee

Reviewer 2 Report
Khoon and coworkers reported the free-radical polymerization for grapting acrylonitrile on expoxidized natural rubber. The product of the polymers were characterization from different aspects, including NMR, GPC, thermal and electrical analysis. A few minor points shall be addressed before the final acceptance.
- Scheme 1, please add the chemical structure of the product in each reaction.
- Table 1, all the polymers seem to have a large dispersity value. Could the authors analyze why these values are so high and what kind of applications do the author expect from these products? Please include these into the manuscript.
Author Response

(The authors gave the same response as above.)

Reviewer 3 Report
General comments
No supplementary file. Can't completely review this manuscript.
Specific comments
Pretty comprehensive work. Authors have carried out almost all essential characterizations but perhaps GPC data would add more value to the manuscript. However the data presented in this manuscript is enough. But there are some minor niggles which must be rectified before publication.
In abstract, In lines 27 to 29 authors claim that "All in all, the synthesized product shows superior properties against traditional and synthetic rubber on the market and ready for wide applications such as composites, coating, etc." which is pure speculation. Remove this unsubstantiated statement. Besides, there is no "TRADITIONAL" rubber. There are only natural and synthetic rubbers.
Figure 6: Authors must include DMA of ENR25 for comparison.
Manuscript mentions supplementary images in supplementary file. But this reviewer could not find supplementary file accompanying this manuscript. Rectify this anomaly.
In line 512, authors compared their DMA data with Ramesan & Alex[43]. But Acrylonitrile‐co‐butadiene rubber used by Ramesan and Alex was Aparene N553 NS having 34 % bound acrylonitrile content which is substantially higher when compared to AN in this work. Justify this comparison.
Besides there are some minor niggles, but the fact that supplementary file is absent makes it difficult to completely review this manuscript.
Author Response

(The authors gave the same response as above.)
